# Determinants of E-Cigarette and Cigarette Use among Youth and Young Adults in Poland—PolNicoYouth Study

**DOI:** 10.3390/ijerph191811512

**Published:** 2022-09-13

**Authors:** Ilona Wężyk-Caba, Małgorzata Znyk, Radosław Zajdel, Łukasz Balwicki, Anna Tyrańska-Fobke, Grzegorz Juszczyk, Karolina Zajdel, Beata Świątkowska, Dorota Kaleta

**Affiliations:** 1Department of Hygiene and Epidemiology, Medical University of Lodz, 7/9 Żeligowskiego Street, 90-752 Łódź, Poland; 2Department of Business and Informatics, University of Łódź, POW 3/5 Street, 90-255 Łódź, Poland; 3Department of Public Health and Social Medicine, Medical University of Gdańsk, 7 Dębinki Street, 80-210 Gdańsk, Poland; 4Department of Public Health, Medical University of Warsaw, 5 Jana Nielubowicza St., 02-097 Warsaw, Poland; 5Department of Medical Informatics and Statistics, Medical University of Lodz, 1 Hallera Sq., 90-645 Łódź, Poland

**Keywords:** young adults, e-cigarette use, predictors of smoking

## Abstract

Teen use of tobacco-related products is a significant public health concern. This study evaluated the predictors of e-cigarette use among secondary school students who were never cigarette smokers and ever cigarette smokers in Poland. Methods: This study examined a sample of Polish youths aged 13–19 (*n* = 19,241) attending 200 schools, 12 on average in each county. The study was a part of the National Health Program in Poland for 2016–2020. Logistic regression and multivariable logistic regression models were used to calculate crude and adjusted odds ratios. Results: Of all participants, 32.5% were ever cigarette users. Among the never cigarette users, 13.6% were deemed susceptible to e-cigarette use. Among the ever cigarette users, 60.6% were deemed susceptible to e-cigarette use. Of those susceptible to e-cigarette use, 68.2% were among the 32.5% ever cigarette users. The profile of e-cigarette use among never e-cigarette users also included: pocket money available per month (more than 150 PLN) (OR = 1.7; *p* = 0.001), 16–17 years old (OR = 1.9; *p* = 0.001), parental tobacco smoking and e-cigarette usage (OR = 2.0; *p* = 0.01 and OR = 1.7; *p* = 0.001 respectively), maternal secondary education (OR = 1.1; *p* = 0.04), and living in big cities >500,000 inhabitants (OR = 1.4; *p* = 0.04). E-cigarette users among ever cigarette users were similar to never cigarette users in their opinion that e-cigarette use is less harmful than traditional smoking (OR = 1.6; *p* = 0.0012) and living with both parents smoking cigarettes (OR = 1.3; *p* = 0.02). Additionally, the determinants were: female gender (OR = 1.5; *p* = 0.009) in the age group less than 15 years of age (OR = 1.3; *p* = 0.007). Conclusions: The major determinant of e-cigarette use in this population was prior smoking. Additionally, the results revealed that fairly obvious predictors such as parental smoking and a belief in the less harmfulness of e-cigarette use are important determinants for smoking among never or ever e-cigarette users.

## 1. Introduction

Cigarette smoking causes more than 8 million deaths worldwide every year (1.2 million from passive smoking and 7 million from active smoking), of which 7 million are related to active smoking and 1.2 million to passive smoking [1]. In Poland, tobacco use is responsible for around 70,000 deaths per year [2]. In recent years, the range of tobacco products introduced to the market has expanded, while the decline in the use of tobacco has not been observed. Alternative forms of nicotine delivery, such as electronic cigarettes (e-cigarette use) and heated tobacco products or HTPs have gained popularity [3,4]. Additionally, the phenomenon of dual use of tobacco and e-cigarette is also observed [2].

Estimated data on the number of tobacco smokers place Poland slightly above the European average. Over the last few years, an alarming halt in the downward trend in the percentage of smokers has been observed. The use of electronic devices or heated tobacco likely contributes to the increase in the percentage of people using nicotine products [5].

E-cigarette use has been introduced into the market to help reduce or quit smoking and is gaining popularity around the world, especially among young people, despite the potential negative health effects of its use [6,7]. E-cigarettes, while not risk-free, are much less dangerous than cigarettes.

In Poland, 22% of adolescents reported that they had ever used an e-cigarette and 27% indicated that they had used an e-cigarette in the last month [8]. Current e-cigarette use among teenagers increased from 5.5% in 2010–2011 to 29.9% in 2013–2014. Dual use of tobacco and e-cigarette was also significantly higher (3.6% and 21.8%, respectively) [7,8]. Research conducted among young adults from rural areas in Poland in a disadvantaged socioeconomic situation also shows a high percentage of young people susceptible to using an e-cigarette (78% of ever and 68% of never e-cigarette users) [9].

The increasing popularity of e-cigarettes among young people is related to curiosity, flavoring/taste, wide promotion and advertising, the attractiveness of the product itself, and its low perceived harm compared to other tobacco products [10,11,12]. E-cigarette use may indicate young people’s independence and attractiveness [13,14,15,16]. Many researchers say that the most common reason for young adults’ experimenting with e-cigarette use is curiosity and an irresistible desire to try something new [17,18,19,20].

The three main reasons for experimenting with e-cigarettes given by e-cigarette users among adolescents, regardless of school level and smoking status, are curiosity, the availability of attractive flavors, and the influence of friends [11]. Among the most frequently mentioned reasons for e-cigarette use in Poland, respondents mention the fashion for vaping (28%), the desire to take care of their health (17%), and saving money (14%), while 22% indicate other reasons for e-cigarette use, such as curiosity [21,22].

E-cigarette use is perceived as “cool’’ and their appearance and design are considered attractive [11]. The innovative features of e-cigarette use are particularly attractive to teens and young adults, affecting their susceptibility to using these products. Some researchers have shown that attractive flavors induce teens to try e-cigarettes [11,23]. Sweet flavors (e.g., fruit, candy) are the most often preferred and chosen by adolescents [24,25].

Exposure to e-cigarette advertising, the low cost of buying them, the ability to e-cigarettes anywhere, and the desire to quit smoking traditional cigarettes are important determinants of the global e-cigarette use among young adults [26,27].

In Poland, since the launch of e-cigarettes marketing in 2008, legal regulations for a long time did not apply to the age limit for e-cigarette use and purchase. This made them easily accessible to young people [28]. Promotion, advertising (e.g., on the Internet, in the press, television/cinema, in shops), and sponsorship of e-cigarette was legal, and could contribute, similar traditional smoking, to the initiation of smoking among young people. In 2016, the amendment of the TCA specified the minimum age for the purchase of e-cigarette use (18 years of age), and banned the promotion, advertising, and sponsorship of these products to the same extent as for traditional cigarettes [29].

Young people are less aware of the health risks of e-cigarette use and are more susceptible to their use than adults [30,31]. Moreover, research shows that adolescents and young adults who use e-cigarettes are more likely to use alcohol and cannabis [32,33].

The aim of this study was to evaluate the predictors of e-cigarette use among secondary school students who were never cigarette smokers and ever cigarette smokers in Poland. The study is based on a representative national sample of young people aged 13–19 years old from each of the counties in Poland. Although many studies have been published on this topic, there is still a need to identify the predictors of e-cigarette use in the specific environment (national study) to adjust the intervention to the needs of the specific population (country specific) and develop a health program based on the knowledge of the specificity of the population (legislation, social norms, and characteristics of the population).

## 2. Materials and Methods

### 2.1. Study Sample

This cross-sectional study was carried out in January and February 2020 among 19,241 secondary school students aged 13–19 years old from 200 Polish schools (grammar school (four-year general secondary school completed with the matriculation examination), technical secondary school (five-year school with a vocational profile completed with the matriculation examination) and vocational secondary school (three-year school)), which represents 1.71% of the relevant population. The selection of units for the study was nationwide, stratified–random in the county, divided into urban and rural areas and school type. A random selection of test units was used. Selection of the units was based on a register of educational institutions, containing 5523 records, created by the Authors. The target respondents were students from classes in institutions selected for the study. All of the 200 schools, an average of 12 schools in each county selected for the project, agreed to participate. The response rate was 98.5%.

The sample covered from 0.84 to 4.14% of the population (students enrolled in secondary schools) per county and allowed for confidence in the results at the level of 95%, fraction size 0.5%, and a maximum error of 4% at the level of each county.

All surveys completed were conducted using the CAWI (Computer-Assisted Web Interview) technique. The study was carried out using SURNEO software. For the purposes of this project, 3 call center stations were launched. The study was performed by an external company. Parents’ consent to participate in the study of children was not required as the students completed the questionnaire during school activities. Despite prior consent from the school management to participate in the study, additional consent was obtained each time students participated in the study. The average time to complete the form was around 10 min.

The study was a part of the national program for combating health consequences of using tobacco and related products, financed by the National Health Program, Ministry of Health in Poland. The National Institute of Public Health PZH—National Research Institute Bioethical Committee Board approved the study (resolution number 3/2019; 13 November 2019).

### 2.2. Questionnaire

The CAWI online interview consists in completing the online version of the questionnaire by the respondent working on their own. The link to the questionnaire was made available to all the respondents. The respondents answered the questions in order, and the software included logical checks of the responses where possible. The respondents could not move on to the next question if they did not answer the previous one, marked more answers than were permitted, or chose mutually exclusive answers. This allowed for a level of quality control in the data collection process despite the questionnaire being self-completed by the respondents. This increased the reliability of the data collection by avoiding mistakes that may occur when encoding data independently or during data entry.

The interview included questions about demographics, socioeconomic status, lifestyle factors, and the main questionnaire adopted from the Global Youth Tobacco Survey (GYTS) recommended by the World Health Organization (WHO) and the Center for Diseases Control and Prevention (CDC) used for systematic monitoring of youth tobacco use [34].

E-cigarette use was coded on the basis of the answers to the following standardized questions adapted from Pierce et al.: (1) “If one of your friends offered you an e-cigarette, would you try it?”, (2) “At any time during the next 12 months, do you think you will use an e-cigarette?” [35]. The students were categorized as nonusers if they answered “definitely not” to both questions. Other combinations of the answers (“probably not”, “probably yes”, “definitely yes”) were considered as indicating the use of an e-cigarette.

In the case of the variable “Relative harmfulness of e-cigarettes compared to traditional cigarettes”, a comparison of perceived relative harmfulness in relation to traditional cigarettes was used.

On the other hand, in the case of the “Exposure to tobacco marketing” variable, the marketing exposures faced by the surveyed youth in the last 30 days in a shop, a disco, and on the Internet were analyzed. It is worth pointing out here that advertising, promotion, and sponsorship of tobacco and e-cigarettes are completely banned in Poland.

Four groups of respondents were distinguished in the sample: never e-cigarette users, never cigarette users (to verify e-cigarette use), ever e-cigarette users and ever cigarette users. The never e-cigarette or cigarette users were the respondents who answered “no” to the question “Have you ever (even once) used cigarettes/e-cigarettes?”. Those who answered “yes” to that question but declared cigarettes or e-cigarette not used in the past 30 days were classified as ever users.

Sociodemographic characteristics (gender, age, school grade, money available per month, father’s and mother’s education) were collected together with variables related to smoking of traditional cigarettes and the use of e-cigarettes (smoking status, parental and friends’ smoking and e-cigarette use, perception of attractiveness and harmfulness of the products).

### 2.3. Statistical Analysis

The analyses were conducted using STATISTICA Windows XP version 12.0 (StatSoft Poland Inc., Tulsa, OK, USA). First, the descriptive statistics and distribution of the study variables were calculated. To identify predictors of e-cigarette use, univariate and multivariate logistic regression analyses (with the results presented as odds ratio (OR) and 95% confidence intervals (95% CI)) were performed. The variables with *p*-values of 0.1 or less from the univariate analysis were included in the multivariate model. Age, as a factor strongly correlated with school grade, was not included in the final analysis. A *p*-value below 0.05 was considered statistically significant.

## 3. Results

### 3.1. Characteristics of the Study Population

The study population consisted of 19,241 pupils, aged 13–19 years old, from secondary schools in Poland. The sample included 52.37% male participants (Table 1).

Most of the respondents attended vocational or technical schools (56.59%), while 43.41% attended a grammar school. The largest groups of the respondents lived in rural areas (47.99%) and small towns with <20,000 inhabitants (20.70%), whereas big cities > 500,000 inhabitants were the place of residence for 4.03% of respondents.

Almost 22% of the respondents were ever e-cigarette users, whereas 30% were current e-cigarette users. In the surveyed sample, 59.17% of the respondents stated that they were not exposed to tobacco marketing. Only 24.26% of the surveyed students believed that e-cigarette use was similarly harmful as traditional cigarettes. Most of the students’ parents did not use tobacco products (58.12%) in the case of traditional cigarettes, 94% for e-cigarette use, and 96% for heated tobacco. Most mothers of the respondents had higher education (45%) whereas fathers had mostly secondary education (42%). Fifty-nine percent of the respondents declared having less than 150 PLN per month of pocket money.

### 3.2. E-Cigarette Use

In the study population, there were 9313 never cigarette users and 4488 ever cigarette users, totaling 13,801 included in the analysis. Of these 13,801, 4488 (32.5%) were ever cigarette users. Among the never cigarette users, 1271 (13.6%) were deemed susceptible to e-cigarette use. Among ever cigarette users, 2720 (60.6%) were deemed susceptible to e-cigarette use. The predictors of e-cigarette use were assessed among never cigarette and never e-cigarette users.

The others predictors of e-cigarette use among never cigarette smokers from univariate analysis were: age group 16–17 (OR = 1.69; *p* = 0.001), big city inhabitants (>500,000 inhabitants) (OR = 1.83; *p* = 0.04), more than 150 PLN pocket money per month (OR = 1.45; *p* = 0.01), maternal secondary education (OR = 1.2; *p* = 0.01), parental traditional or e-cigarette use (OR = 1.30; *p* = 0.001 or OR = 4.56; *p* = 0.04, respectively), and opinion that e-cigarette use is less harmful than traditional cigarettes (OR = 1.60; *p* = 0.03) (Table 2).

The multivariate logistic regression analysis results are similar to univariate analysis. According to the results of the adjusted analysis the profile of e-cigarette use among never e-cigarette users were as follows: pocket money available per month (more than 150 PLN) (OR = 1.71; *p* = 0.001), age 16–17 (OR = 1.93; *p* = 0.001), parental tobacco smoking and e-cigarette use (OR = 2.03; *p* = 0.01 and OR = 1.70; *p* = 0.001, respectively), maternal secondary education (OR = 1.14; *p* = 0.04), and living in big cities (OR = 1.48; *p* = 0.04) (Table 2).

The univariate analysis indicated that determinants of e-cigarette use among ever cigarette users were: female gender (OR = 1.27; *p* < 0.001), less than 15 years of age (OR = 1.26; *p* = 0.019), lower school grade (OR = 1.26; *p* = 0.001), parental e-cigarette use (OR = 1.36; *p* = 0.001), and opinion that e-cigarette use is less harmful than traditional smoking (OR = 1.31; *p* = 0.003) (Table 3).

The profile of e-cigarette users, according to multivariate logistic regression analysis, included female gender (OR = 1.50; *p* = 0.009), in the age group less than 15 years of age (OR = 1.37; *p* = 0.007), living with parents who smoked cigarettes (OR = 1.39; *p* = 0.02), and thinking that e-cigarette use is less harmful than traditional smoking (OR = 1.69; *p* = 0.0012) (Table 3).

As shown by the data in Table 2 and Table 3, 68.2% of those susceptible to e-cigarette use were among the 32.5% ever cigarette users. For this group, the OR for the smoking variable is approximately 5.0. Additionally, e-cigarette users among ever cigarette users were similar to never cigarette users in their opinion that e-cigarette use is less harmful than traditional smoking (OR = 1.6; *p* = 0.0012 for ever cigarette users and OR = 1.6; *p* = 0.03 for never cigarette users) and living with both parents smoking cigarettes (OR = 1.3; *p* = 0.02 for ever cigarette users and OR = 1.3; *p* = 0.02 for never cigarette users).

## 4. Discussion

The current study is one of the first based on a representative, national sample of young people in Poland concerning predictors of e-cigarette use, such as prior smoking as a major factor, and also age group, place of living, pocket money per month, mother’s education, parental smoking, relative harmfulness, and e-cigarette/traditional cigarette assessment.

A common liability to addiction (CLA) says that teens who are likely to experiment with one prohibited substance are more likely to experiment with other such substances, compared to teens who are not susceptible to such experiments. The data from the research are in line with the common liability model, where people prone to nicotine use are more likely to use both cigarettes and e-cigarettes [36]. The profile of e-cigarette use, however, is different in different countries, so there is a need to adjust interventions based on the social influences, population characteristics, and legislation specific to a particular area.

The fact that in our study the major predictor of e-cigarette use is prior smoking suggests that e-cigarettes divert smokers away from cigarettes. In some cases the e-cigarette can be used to help reduce or quit smoking traditional cigarettes, despite the potential negative health effects of their use [6,7]. Our analysis indicates that the predictors of e-cigarette use among never cigarette smokers were as follows: money available per month (more than 150 PLN), age (16–17), parental tobacco smoking and e-cigarette use, maternal secondary education, and living in big cities. Predictors of e-cigarette use among ever cigarette smokers were prior smoking, living with parents who both smoke cigarettes, and the opinion that e-cigarette use is less harmful than traditional smoking. This last predictor is also similar to never cigarette smokers. Additional determinants included the female gender in the age group less than 15 years of age. Polish youth is at the forefront of the European Union where e-cigarette use is increasing (from 6% in 2011 to 29.9% in 2014) [35]. The Global Youth Tobacco Survey (GYTS) showed the highest prevalence of the current e-cigarette use in Poland (2016—23.4%) as well as in Ukraine (2017—8.4%) and Latvia (2019—18.0%) [37]. The ESPAD (European School Survey Project on Alcohol and Drugs) study in 2019 showed that e-cigarette use was more prevalent among the older group of adolescents aged 17–18 years than among the 15–16 age group regarding the last 30 days before the test (36.5.% and 30.3%, respectively) and ever use (64.8.% and 56.3%) [38]. Reports from the UK showed that between 1% and 3% of young people aged 11–16 used e-cigarettes regularly, and between 7% and 18% used them frequently [39]. Studies in the United States found that 4.3% of middle school students and 11.3% of high school students reported e-cigarette use [40].

As mentioned above, in our study the major predictor of e-cigarette use is prior smoking, but our results also showed that female gender and younger age were associated with e-cigarette use, especially among ever cigarette smokers, whereas for example, in another study the susceptible youth were more likely to be older and male [41]. This confirms the suggestion of country-specific differences. To summarize, it is important to continue international research, taking into account the search for cultural differences and other factors influencing the perception and use of tobacco products by young people.

A study of Canadian youths showed that regularly seeing people’s e-cigarette use is associated with initiating e-cigarette use among youth and young adults [42]. Kwon et al. 2018 found that the exposure to environmental tobacco smoke (e.g., smoking parents) increased the susceptibility to e-cigarette use [43]. Additionally, in the study performed in rural areas in Poland, smoking cigarettes by friends or parents was a predictor of e-cigarette use [9]. This is consistent with our results regarding parental smoking.

A secondary level of maternal education was also a predictor of e-cigarette use among socioeconomically disadvantaged youths from rural areas of Poland [9], which is in line with the findings from our study.

The results of our study indicate that the higher amount of spending money available per month was associated with e-cigarette use. The observed relationship between more pocket money and e-cigarette use may be due to the fact that adolescents with more spending money can afford to buy e-cigarettes [44,45]. Caregivers who provide pocket money to young people should pay attention to how the money is spent. However, in a study by Kaleta et al. conducted in Poland, adolescents who had more money per month were less susceptible to experiments with e-cigarette use [9]. In a study covering people over 15 years old, 14% of respondents in Poland indicated saving money as the reason for using e-cigarettes, in addition to the desire to look after their health (17%) [21]. The high cost of the product is, on the other hand, the reason why it is difficult for young people in the Russian Federation to access e-cigarettes [46].

Numerous studies of young adults on e-cigarette use have been conducted in larger urban centers [47,48]. In the study by Park et al., e-cigarette use was associated with a higher average weekly allowance and residence in urban areas [49]. Similarly, in our study, living in large cities was a predictor of e-cigarette use. In contrast, in the study by Kaleta et al., a high percentage of young people from socially disadvantaged rural areas in Poland were e-cigarette users [8].

Our study also found that thinking of e-cigarette use as less harmful is related to e-cigarette use. The perceptions of e-cigarette use vary significantly among countries. Different attitudes and perceptions of e-cigarette use among young adults may result from cultural differences and the shape of the anti-smoking policy in each country [50]. While e-cigarette use is not risk-free, there are good reasons to believe that they are much less dangerous and possibly less addictive than cigarettes [50,51].

Research shows that e-cigarette use is less harmful to health than traditional cigarettes [29,52,53,54]. The belief that e-cigarette use is less harmful than traditional cigarettes is a risk factor for starting e-cigarette use [9]. Adolescents who had ever used e-cigarettes saw them as less harmful than traditional cigarettes to a greater degree compared to those who had never used e-cigarettes [55,56].

E-cigarette use-only perceived traditional cigarettes as more harmful than the group of dual users [13]. In other studies, perceiving e-cigarette use as addictive and harmful acted as protective factors against use [45]. Many young adults support the view that e-cigarette use is less harmful than traditional cigarette use [56,57].

The present study is the most recent report on the determinants of e-cigarette use among young people in Poland. The study population is a representative national sample of young people aged 13–19 years old from each of the counties in Poland, so the results can be generalized to the entire population of young people in Poland. The study protocol and questionnaire were based on the standardized tools used in GYTS.

The study had some limitations. The first was the cross-sectional design, unable to establish causation between the determinants of e-cigarette use and actual use. The changes over time cannot be thus predicted. Additionally, the analysis did not control for other substances such as drugs or alcohol.

Despite the limitations, the current study provides information about the predictors of e-cigarette use among Polish youths. It is worth pointing out, however, that further research on this topic is necessary to help guide tobacco policy in Poland. It is also important to continue research among Polish youth, taking into account in particular the place of residence, in order to more clearly identify the cause-and-effect relationships of the phenomenon of using tobacco products by young people.

Restrictions on the sale of e-cigarettes in Poland introduced recently, e.g., purchase of e-cigarettes only by adults, limiting the availability of the product by banning online trade, and competitive prices compared to traditional cigarettes, may limit the use of e-cigarettes. In addition, pending the results of further research in Poland, it seems reasonable to discourage all use of tobacco and cigarette-like products (e.g., marijuana).

## 5. Conclusions

The major determinant of e-cigarette use in this population was prior smoking. Additionally, the results revealed that such fairly obvious predictors as parental smoking and a belief in the less harmfulness of e-cigarette use are important determinants of smoking among never or ever e-cigarette users. Based on this finding, intervention should be focused on educational campaigns targeting young people and their parents with information on the addictiveness and harmful effects of e-cigarette use, cigarettes, and cigarette-like products, and should address peer use and perceptions of harm to prevent the uptake of e-cigarettes by young people. However, it is essential that education campaigns should not only provide information about known harmful effects of using all tobacco-related and cigarette-like products, but also teach teens and young adults effective strategies to cope with peer pressure.

## Figures and Tables

**Table 1 ijerph-19-11512-t001:** Characteristics of the study population (N = 19,241).

Characteristics			N (%)
All			19,241 (100)
Gender	Female		9164 (47.63)
	Male		10,077 (52.37)
Age	≤15		3634 (19.10)
	16–17		9877 (51.92)
	≥18		5511 (28.97)
Type of school	Grammar school		8213 (43.41)
	Vocational/technical		10,708 (56.59)
Mother’s education	Primary/vocational		2377 (13.12)
	Secondary education		5736 (38.90)
	Higher education		6632 (44.98)
Father’s education	Primary/vocational		3442 (24.20)
	Secondary education		6022 (42.33)
	Higher education		4761 (33.47)
Pocket money per month (PLN)	≤150		8796 (58.63)
	>150		6206 (41.37)
Parental smoking	None		10,297 (58.12)
	Mother		1772 (10.00)
	Father		3150 (17.78)
	Both		2499 (14.10)
Parental e-cigarette use	None		16,431 (93.97)
	Mother		275 (1.57)
	Father		542 (3.10)
	Both		237 (1.36)
Parental heated tobacco product use	None		16,476 (96.21)
	Mother		204 (1.19)
	Father		221 (1.29)
	Both		224 (1.31)
Relative harmfulness of e-cigarettes compared to traditional cigarettes	Far less harmful		3427 (18.27)
	Less harmful		4724 (25.19)
	Similar harmful		4551 (24.26)
	More harmful		1939 (10.34)
	Far more harmful		1056 (5.63)
	I do not know		3060 (16.31)
Place of living	Rural		8771 (47.99)
	<20,000 inhabitants		3784 (20.70)
	20–99,000 inhabitants		3142 (17.20)
	100–500,000 Inhabitants		1841 (10.07)
	>500,000 inhabitants		737 (4.03)
Exposure to tobacco marketing	Yes		7857 (40.83)
	No		11,384 (59.17)
Cigarette smoking status	Never		9313 (48.4)
	Ever		9928 (51.6)
	Current		4488 (23.3)
E-cigarette use status * *n* = 19,113 (100.00)	Never	Yes	9126 (47.75)
		No	
	Ever	Yes	9987 (52.25)
		No	
	Current	Yes	5817 (30.43)
		No	

* ever e-cig user—surveyed adolescents who have ever tried e-cigarette use; current e-cig user—surveyed adolescents who e-cigarette used at least once during the last 30 days.

**Table 2 ijerph-19-11512-t002:** The predictors of e-cigarette use among never cigarette smokers.

Characteristics		Never Cigarette Users	E-Cigarette Users	OR Crude	95% CI	*p*		OR Adjusted	95% CI	*p*
All		9313 (100.00)	1271							
Gender	Female	4475 (48.05)	378 (71.05)	Ref.				Ref.		
	Male	4838 (51.95)	543 (73.48)	1.13	0.88–1.45	0.34		1.08	0.88–1.32	0.49
Age	≤15	2398 (26.16)	221 (69.94)	1.26	0.88–1.81	0.216		1.23	0.99–1.53	0.06
	16–17	5001 (54.55)	544 (75.77)	1.69	1.23–2.33	0.001		1.93	1.35–2.76	0.001
	≥18	1769 (19.30)	148 (64.91)	Ref				Ref.		
Type of school	Grammar school	4455 (48.64)	437 (70.94)	1.15	0.90–1.47	0.260		1.14	0.86–1.50	0.37
	Vocational/ technical	4704 (51.36)	481 (73.77)	Ref.				Ref.		
Mother’s education	Primary/vocational	1051 (15.02)	108 (70.13)	Ref.			Ref.			
	Secondary education	2653 (37.92)	298 (71.98)	1.20	1.13–1.98	0.01		1.14	1.04–1.54	0.04
	Higher education	3292 (47.06)	343 (71.16)	1.05	0.1–1.56	0.806		1.05	0.85–1.28	0.67
Father’s education	Primary/vocational	1561 (22.98)	156 (68.12)	Ref.				Ref.		
	Secondary education	2847 (41.90)	315 (72.92)	1.26	0.89–1.79	0.20		1.24	0.95–1.64	0.12
	Higher education	2386 (35.12)	259 (74.43)	1.36	0.94–1.97	0.10		1.24	0.99–1.56	0.06
Pocket money per month (PLN)	≤150	2532 (36.03)	249 (68.03)	Ref.				Ref.		
	>150	4496 (63.97)	526 (75.47)	1.45	1.09–1.91	0.01		1.71	1.25–2.34	0.001
Parental smoking	None	5650 (66.27)	577 (72.13)	Ref.				Ref.		
	Mother	705 (8.27)	93 (77.50)	1.531	0.91–2.58	0.108		1.24	0.99–1.56	0.06
	Father	1349 (15.82)	144 (69.23)	1.15	0.825–1.604	0.410		1.54	0.79–3.00	0.20
	Both	822 (9.64)	70 (74.47)	1.30	1.75–2.25	0.001		2.03	1.45–3.18	0.01
Parental e-cigarette use	None	8138 (95.68)	842 (71.78)	Ref				Ref.		
	Mother	93 (1.09)	14 (87.50)	1.33	0.26–1.66	0.21		1.39	0.27–3.59	0.36
	Father	213 (2.50)	34 (85.00)	1.89	0.69–5.33	0.38		1.58	0.78–6.28	0.78
	Both	61 (0.72)	1 (33.33)	4.56	1.08–19.33	0.04		1.70	1.30–2.24	0.001
Parental heated tobacco product use	None	8089 (97.73)	849 (72.01)	Ref.				Ref.		
	Mother	63 (0.76)	7 (77.78)	1.59	0.28–3.25	0.44		1.78	0.55–4.21	0.69
	Father	69 (0.83)	7 (77.78)	1.46	0.41–5.19	0.56		1.11	0.44–2.80	0.83
	Both	56 (0.68)	3 (75.00)	0.85	0.22–3.31	0.82		1.41	0.98–2.04	0.07
Relative harmfulness of e-cigarettes compared to traditional cigarettes	Far less harmful	1135 (12.48)	169 (86.22)	Ref.				Ref.		
	Less harmful	2221 (24.42)	354 (79.02)	1.60	1.38	1.96	0.03	1.85	0.86–1.97	0.34
	Similar harmful	2481 (27.28)	186 (62.84)	0.27	0.17	1.432	0.29	0.32	0.21–1.47	0.25
	More harmful	933 (10.26)	47 (53.41)	0.18	0.10	1.328	0.78	0.36	0.23–1.56	0.67
	Far more harmful	519 (5.71)	18 (50.00)	0.16	0.07	1.345	0.32	0.19	0.11–1.32	0.45
	I do not know	1805 (19.85)	145 (71.78)	0.41	0.24	1.676	0.45	0.32	0.21–1.50	0.84
Place of living	Rural	4431 (50.15)	423 (72.06)	Ref.				Ref.		
	<20,000 inhabitants	1768 (20.01)	172 (70.49)	0.93	0.67	1.29	0.65	1.24	0.95–1.64	0.12
	20–99,000 inhabitants	1437 (16.26)	165 (73.01)	1.05	0.74	1.48	0.79	1.11	0.84–1.47	0.48
	100–500,000 inhabitants	886 (10.03)	108 (75.52)	1.20	0.79	1.82	0.41	1.41	0.96–2.06	0.08
	>500,000 inhabitants	313 (3.54)	33 (82.50)	1.828	1.03	2.214	0.04	1.48	1.32–2.84	0.04
Exposure to tobacco marketing	Yes	3218 (34.55)	355 (73.80)	Ref.				Ref.		
	No	6095 (65.45)	566 (71.65)	0.90	0.70	1.16	0.40	0.88	0.70–1.16	0.41

OR—odds ratio.

**Table 3 ijerph-19-11512-t003:** The predictors of e-cigarette use among ever cigarette smokers.

Characteristics		Ever Cigarette Users	Cigarette Users	OR Crude	95% CI	*p*	OR Adjusted	95% CI	*p*
All		4488 (100.00)	2720 (60.61)						
Gender	Female	2137 (47.62)	1359 (63.59)	1.27	1.13–1.43	<0.001	1.50	1.11–2.03	0.009
	Male	2351 (52.38)	1361 (57.89)	Ref.					
Age	≤15	606 (13.58)	389 (64.19)	1.26	1.04–1.53	0.019	1.37	1.08–1.66	0.007
	16–17	2201 (49.31)	1341 (60.93)	1.10	0.96–1.25	0.17	0.95	0.82–1.09	0.44
	≥18	1657 (37.12)	973 (58.72)	Ref.			Ref.		
Type of school	Grammar school	1862 (41.96)	1187 (63.75)	1.263	1.12–1.43	<0.001	1.38	0.78–1.72	0.21
	Vocational/ technical	2576 (58.04)	1499 (58.19)	Ref.			Ref.		
Mother’s education	Primary/vocational	610 (16.94)	349 (57.21)	Ref.			Ref.		
	Secondary education	1455 (40.41)	878 (60.34)	1.14	0.94–1.38	0.19	1.14	0.92–1.42	0.25
	Higher education	1536 (42.65)	949 (61.78)	1.21	0.99–1.46	0.05	1.10	0.95–1.27	0.19
Father’s education	Primary/vocational	905 (26.10)	523 (57.79)	Ref.					
	Secondary education	1511 (43.58)	915 (60.56)	1.121	0.95–1.33	0.18	1.15	0.95–1.40	0.16
	Higher education	1051 (30.31)	640 (60.89)	1.137	0.95–1.36	0.16	1.23	0.89–1.52	0.12
Pocket money per month (PLN)	≤150	2123 (58.13)	1286 (60.57)	Ref.			Ref.		
	>150	1529 (41.87)	914 (59.78)	1.034	0.90–1.18	0.63	1.09	0.87–1.36	0.45
Parental smoking	None	2423 (57.08)	1499 (61.87)	Ref.			Ref.		
	Mother	439 (10.34)	247 (56.26)	1.080	0.84–1.38	0.539	1.09	0.84–1.41	0.53
	Father	774 (18.23)	453 (58.53)	1.185	0.96–1.47	0.120	1.19	0.25–1.78	0.44
	Both	609 (14.35)	331 (54.35)	1.363	1.14–1.63	0.001	1.39	1.09–1.78	0.02
Parental e-cigarette use	None	3967 (94.27)	2353 (59.31)	Ref.			Ref.		
	Mother	56 (1.33)	36 (64.29)	1.24	0.71–2.14	0.45	1.44	0.70–2.99	0.32
	Father	147 (3.49)	98 (66.67)	1.37	0.97–1.94	0.08	1.28	0.78–2.11	0.25
	Both	38 (0.90)	27 (71.05)	1.68	0.83– 3.40	0.15	1.45	0.89–3.21	0.36
Parental heated tobacco product use	None	4004 (96.93)	2378 (59.39)	Ref.			Ref.		
	Mother	45 (1.09)	33 (73.33)	1.88	0.97–3.65	0.06	1.89	0.58–2.17	0.23
	Father	54 (1.31)	36 (66.67)	1.37	0.77–2.42	0.28	1.25	0.87–2.52	0.38
	Both	28 (0.68)	20 (71.43)	1.71	0.75– 3.89	0.20	1.54	0.78–3.17	0.47
Relative harmfulness of e-cigarettes compared to traditional cigarettes	Far less harmful	941 (21.29)	559 (59.40)	Ref.			Ref.		
	Less harmful	1264 (28.60)	830 (65.66)	1.31	1.10–1.56	0.003	1.69	1.51–1.92	0.012
	Similar harmful	1000 (22.63)	569 (56.90)	0.90	0.75–1.08	0.26	0.85	0.30–2.40	0.57
	More harmful	408 (9.23)	242 (59.31)	0.10	0.79–1.26	0.98	1.11	0.78–3.24	0.15
	Far more harmful	187 (4.23)	91 (48.66)	1.65	0.47–1.89	0.56	1.17	0.99–1.37	0.06
	I do not know	619 (14.01)	371 (59.94)	1.02	0.83–1.26	0.84	1.32	0.78–2.12	0.63
Place of living	Rural	2039 (47.25)	1224 (60.03)	Ref.			Ref.		
	<20,000 inhabitants	922 (21.37)	551 (59.76)	1.03	0.85–1.26	0.73	1.12	0.78–2.14	0.75
	20–99,000 inhabitants	782 (18.12)	461 (58.95)	1.05	0.88–1.24	0.60	1.25	0.96–1.60	0.11
	100–500,000 inhabitants	436 (10.10)	277 (63.53)	1.21	0.95–1.54	0.12	1.50	0.95–2.39	0.08
	>500,000 inhabitants	136 (3.15)	89 (65.44)	1.32	0.90–1.930	0.16	1.10	0.90–1.33	0.33
Exposure to tobacco marketing	Yes	1980 (44.12)	1214 (61.31)	Ref.			Ref.		
	No	2508 (55.88)	1506 (60.05)	1.054	0.94–1.19	0.39	0.96	0.83–1.11	0.55

OR—odds ratio.

## Data Availability

Program zwalczania następstw zdrowotnych używania wyrobów tytoniowych i wyrobów powiązanych w ramach Narodowego Programu Zdrowia Ankietowe badanie młodzieży. Available online: https://www.pzh.gov.pl/wp-content/uploads/2020/06/RAPORT-TYTOŃ-MŁODZIEŻ-GRUDZIEŃ-2019-WERSJA-FINALNA-www.pdf (accessed on 20 December 2021).

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
