# Peer review of "Determinants of E-Cigarette and Cigarette Use among Youth and Young Adults in Poland—PolNicoYouth Study"

_ijerph, 2022, doi:10.3390/ijerph191811512_

Round 1
Reviewer 1 Report (Previous Reviewer 2)
In some ways, this draft, received by this reviewer 21 July 2022, was worse than previous drafts. It is certainly not suitable for publication in its current form, although it contains data worthy of publication.
Apparently in response to comments by this, and possibly other viewers, major statements were added to the abstract and conclusions to the effect that smoking was the major predictor of e-cigarette use and that the predictive factors were the same for use of cigarettes and susceptibility to use e-cigarettes. Unfortunately, the authors did not tabulate the data and provide the narrative in either Results or Discussion sections relative to these two conclusions.
Working from Tables 2 and 3, it appears that there were 9,313 never Cigarette Users and 4,488 ever cigarette users, totaling 13,801 for those included in these two tables. Of these 13,801, 4,488 (32.5%) were Ever Cigarette Users. Among the Never Cigarette Users, 1,271 (13.6%) were deemed susceptible to e-cigarette use. Among Ever Cigarette Users, 2,720 (60.6%) were deemed susceptible to e-cigarette use. Stated another way, 68.2% of those Susceptible to E-cigarette Use were among the 32.5% Ever Cigarette Users. It was based on these data that this reviewer concluded that smoking was the dominant predictive factor. The Odds Ratio for the smoking variable is approximately 5.0, far more than the Odds Ratios provided for the other variables presented in the Abstract, none of which are greater than 1.7.
As to whether the risk factors were the same for smokers and e-cigarette users, any conclusion would require re-tabulation of data from Tables 2 and 3 to address this issue – to identify similarities and differences based on the other factors tabulated in these tables.
Another problem in this paper is that the data in Tables 2 and 3 cannot be tied back to the data in Table 1, and no explanation is given for these differences.
Yet another major problem is that, at the bottom of Table 1, there are more Current E-cigarette Users than there are Ever E-cigarette Users. Here, again, no explanation is provided.
One other item in need of discussion in the Discussion section and reference in the Conclusion and Abstract has to do with the fact that the authors did not tabulate e-cigarette use. They tabulated “susceptibility to e-cigarette use.” This, too, requires explanation, as well as the degree to which “susceptibility” relates to actual use of e-cigarettes, and why the authors chose to use this unusual approach to tabulating and discussing predictors of e-cigarette use.
One small grammatical issue, correct in the prior draft, but in error in this draft is to be found in lines 57 and 58 having to do with placement of the word “used.”
Author Response
Thank you for your valuable comments. Please see the attachment.

Reviewer 2 Report (New Reviewer)
The paper recovers a current and socially relevant topic, the introduction, the method and the results are well written, in the discussion a good link was made between the results and the proposal of specific restrictions in relation to the sale of electronic cigarettes. However, it is necessary:
- it is important that the discussion is reviewed again considering including proposal for future research.
- review the conclusions, since it is pointed out that it is important to implement educational campaigns to provide information on how to handle peer pressure. To manage peer pressure, strategies need to be taught, since providing information only applies to knowing to health effects of cigarettes and new tobacco products.
Author Response
Thank you for your valuable comments. Please see the attachment.

Round 2
Reviewer 1 Report (Previous Reviewer 2)
While this study includes data worthy of publication, the narrative and conclusions represent fixed views of the authors not consistent with the most prominent finding emanating from the data presented. As noted in my previous review of this paper, the major determinant of e-cigarette use in this population is prior smoking. The refusal to place these data in a separate table with separate discussion and their insistence on suggesting e-cigarettes as a gateway to smoking is not supported by the data presented in this study. The fact that the major predictor of e-cigarette use is prior smoking suggests the opposite -- that e-cigarettes divert smokers away from cigarettes. This possiblity is not even referenced in this paper.
IF THE AUTHORS DISAGREE WITH MY CONCLUSION, AS NOTED ABOVE, I WOULD WELCOME THEIR ANALYSIS AND REASONING SHOWING THAT PRIOR SMOKING IS NOT THE MAJOR DETERMINANT OF E-CIGARETTE USE.
Failing that, I recommend that this paper be rejected.
Author Response
Thank you. Please see the attachment.

This manuscript is a resubmission of an earlier submission. The following is a list of the peer review reports and author responses from that submission.
Round 1
Reviewer 1 Report
Comments to Authors
This manuscript has improved slightly and still needs numerous minor corrections. Clarification of study aims is also needed as currently it is different in the abstract, introductions, results, and discussion. Besides, the distribution of the sample according to the outcome variable remained unclear. There are many scientific wording and editing mistakes in the manuscript. These could have been easily corrected if all Authors take the time to read the manuscript thoroughly before submission. Despite, below I provide my general and specific comments again, the latter is far more than a reviewer’s task.
General comments
The clear aim of the manuscript is missing. The assessment of the relationship between the outcome variable (susceptibility to e-cigarette use) and never/ever cigarette use, and never e-cigarette use variables are highly confusing. Clearly and consistently defining the aims of the study would be extremely important. Currently, there are several different aims instead of one clear aim:
- in the Abstract: “This study evaluated the determinants of susceptibility to both e-cigarettes use and traditional cigarettes in secondary school students in Poland.”
- in the Introduction: „The aim of the present study was to evaluate the determinants of susceptibility to e-cigarette use in youth and young adults.”
- in the Results: Table 2 and 3 again represent different aims: „The predictors of susceptibility to e-cigarette use among never cigarette smokers.” and „The predictors of susceptibility to e-cigarette use among ever cigarette smokers”, respectively.
- in the Discussion: „Our analysis indicates that the determinants of susceptibility to e-cigarette use 255 among never e-cigarette users…”
Throughout the manuscript, when the Authors mention the device electronic cigarette, it is abbreviated incorrectly as “e-cigarettes use” (first in line 52). Please delete "use" from this abbreviation and use „e-cigarettes” when you mean the device and “e-cigarette use” or “vaping" when you mean the use of the device. I also recommend not using plural, that is, change “e-cigarettes use” to “e-cigarette use”. Please correct these throughout the manuscript.
Please consider changing “voivodship” to “county” in the whole manuscript. The term “voivodship” is incorrect, it is “voivodeship” correctly, but I think you rather mean “counties” of Poland.
Specific comments
Title
Please consider changing “…e-cigarette use and cigarette use…” to either “…e-cigarette and cigarette use…” or “…e-cigarette use and cigarette smoking…”.
Abstract
Lines 22-24: This sentence is vague. Please reconsider and reword it.
Lines 29-30: “Almost 61% of ever e-cigarette users were susceptible to cigarette 29 smoking, whereas only 14% of never cigarette users were susceptible.” These results were not presented in the Results section. Please include it. Moreover, the second part of the sentence is blurry, please clarify: never cigarette users were susceptible “to what”?
Line 34: please do not use abbreviation like “thou.”.
Line 35: “E-cigarette susceptible persons among ever users” it is unclear what product was ever used by ever users. Please clearly indicate.
Line 39: I suggest changing “lower harmfulness” to “less harmfulness”.
Line 41: I suggest changing “The intervention…” to “Interventions…”.
Introduction
Lines 48-49: “In Poland, tobacco use is responsible for around 70,000 deaths per year [2].” In the
Response to Reviewer 3 Comments (Round 2), the Authors indicated tobacco-related 90,000 death/year in Poland. Moreover, the cited literature seems incorrect. Please revise the number of death and the citation.
Lines 60-61: This is a strong statement without citation: “E-cigarettes, while not risk-free, are much less dangerous than cigarettes.” Please insert citation for this statement.
Lines 62-63: “In Poland, 22% of adolescents reported that they had ever used e-cigarettes and 27% indicated that they had used e-cigarettes in the last month.” In this sentence, current use was higher than ever use. I suspect this is incorrect. Please revise the data.
Line 74: Please change the citation “[18-21-]” to “[18-21]”.
Lines 75-76: It is unclear whether the Authors explain the reasons for e-cigarette use among adolescents and/or adults. Please clarify.
Line 79: I suggest changing “…and saving money (14%). 22% indicate other…” to „…and saving money (14%), while 22% indicate other…”.
Lines 81-82: I think that citation is missing from the end of this sentence. Please include it.
Lines 83-84: Please change “Some research shown…” to „Some researches have shown…”.
Lines 89-90: This statement in this current form is not true, this should be nuanced. Please consider including “for a long time” in the sentence, preferably like “…any legal regulations for a long time,…”.
Line 107: Please change “…published on this topic three is still a need to…” to „published on this topic there is still a need to…”.
Materials and Methods
Lines 116-117: Please change “…which is a 1.71% all population size.” to „which represents 1.71% of the relevant population.”.
Line 124-126: Please delete the single point “.” in the row and edit the text to be continuous, without empty rows.
Lines 159-160: The wording of the question is incorrect. Please change “If one of your friends offered you a e-cigarette use, would you try it?” to “If one of your friends offered you an e-cigarette, would you try it?”.
Lines 160-161: The question is incomplete (“…do you think you will use a e-cigarette or ?”), please amend it at the end.
Lines 165-166: Please consider rewording this variable: “Relative harmfulness assessment e-cigarettes use/traditional cigarettes" is recommended to change for „relative harmfulness of e-cigarettes compared to traditional cigarettes”. Please also include this change in Table 2 and 3. Besides, I propose changing “comparison of harmfulness” to “comparison of perceived relative harmfulness” as participants indicated their subjective perceptions regarding the harmfulness of e-cigarettes.
Lines 167-168: In my opinion, the term “marketing impacts” is meaningless in the context of the “exposure to tobacco marketing” variable. I suggest changing it for “…the marketing exposures faced by…”.
Lines 169-170: I recommend changing “…that tobacco advertising, promotion, and sponsorship e-cigarettes use is completely banned in Poland” to „…advertising, promotion, and sponsorship of tobacco and e-cigarettes are completely banned in Poland”.
Lines 178-183: On the one hand, HTP use is missing from the list of additional included variables. On the other hand, “the ban on smoking at home and school” and “Psychoactive substances (including drugs) consumption by the students…” variables were not included in the current analysis, so I would suggest omitting them.
Results
Subsection 3.1: The proportion of current/ever/never e-cigarette use and cigarette smoking status are unclear. Please clearly describe your outcome variables and indicate the distribution of the sample according to these variables.
Table 1: Data on never, ever, and current smokers in the sample are missing from the table. Please include these essential data. The variable name “Parental heated tobacco” is meaningless, I suggest changing it to “Parental heated tobacco product use” and change it also in Table 2 and 3. Please change the variable name “e-cig user” to “e-cigarette use status”, and at this variable, change “former” users to “ever” users as ever use is one of the main variable of interest in this study. Please consider that generally former users of a product are those who previously used it regularly.
Line 205: Please change “former e-cigarette users” to “ever e-cigarette users”.
Line 208: Please change “…similarly harmful to traditional…” to “…similarly harmful than traditional…”.
Line 210: Please delete “(Table 2)” from the text as the text relates to Table 1.
Subsection 3.2: There is not any data about the proportion of susceptibility for e-cigarette use in the sample. Please include these in this subsection. These data would be essential.
Line 214: The predictors of susceptibility to e-cigarette use were assessed only among never cigarette smokers or both among never cigarette and never e-cigarette users? Please clarify it.
Line 217: Please consistently use “traditional cigarette” or “conventional cigarette”. Now these are interchangeably used throughout the manuscript. “Traditional cigarette” is more often used in the text, so I suggest using this term throughout the manuscript.
Line 218: An “OR” is missing from “…p=0.001 or 4.56; p=0.04…”, please change it to „…p=0.001 or OR=4.56; p=0.04…”.
Table 2 and 3: Regarding “type of school” variable, the Authors have changed the reference category in Table 34 compared to Table 2 and gave no explanation for this. I have noted this in my first review but did not get any response for this so far. In Table 2, please correct the variable “Parental e-ciggarettes” to “Parental e-cigarette use”.
Discussion
Lines 245-246: This first sentence is incomplete as did not explain among whom concerned the predictors of e-cigarette use susceptibility. Please clarify it.
Lines 255-256: “Our analysis indicates that the determinants of susceptibility to e-cigarette use among never e-cigarette users…” such analysis was not conducted. According to Table 2, the Authors analyzed „The predictors of susceptibility to e-cigarette use among never cigarette smokers.” In my comment to Line 214, I have asked whether the predictors were assessed only among never cigarette smokers or both among never cigarette and never e-cigarette users. The assessment of the relationship between the outcome variable and never/ever cigarette use and never e-cigarette use variables are highly confusing. Clearly and consistently defining the aims of the study would be extremely important.
Line 262: Please explain the determinants a bit more detailed (where, which study, what population, when?).
Lines 268-270: Percentages in brackets does not meet with the findings stated previously in the text. Please revise it.
Line 276: „…especially among ever users,…” ever users of what? Please clarify.
Lines 289-290: Delete an Enter and edit the 2nd sentence right after the first sentence. These belong to each other.
Lines 293-295: What could be the reason for different results among adolescents in the same country? Please explain.
Lines 311-312: This is a strong statement without citation. Please insert citation for this statement.
Lines 313-314: These references do not support this strong statement. Please carefully select citations. Delete double points at the end.
Line 323: Please revise this: „…e-cigarettes are use healthier…”.
Lines 338-340: E-cigarettes already have sales restriction in Poland. Please revise the sentence.
Conclusion
Line 343: I suggest changing “lower harmfulness” to “less harmfulness”.
Lines 344-345: “Our results shown that the determinants of e-cigarette use and cigarette smoking are the same.” Such analyses were not included in the manuscript. Only the determinants of e-cigarette use susceptibility. Please revise it.
Lines 352-354: It is a very lay wording. Authors may mean that for adolescent smokers, switching to e-cigarette use cannot be considered as a potential harm reduction method. On the other hand, emphasizing the potential gateway effect of e-cigarettes is also needed.
References
Edit References as some of the items have outlying position (inner than others).
Author Response
The authors really appreciate all your kindly comments. Please see the attachment.

Reviewer 2 Report
Sorry, but this paper is not yet suitable for publication. While most of the changes I recommended have been included, at least one critically important item has not. In addition, while inserting the changes I recommended, the authors introduced a new problem – English grammar having to do with subject/object agreement relative to singular or plural.
The Abstract is a critically important part of the paper. Most people with an interest in the topic covered by the paper will review the abstract and not bother reading the rest of the paper.
The abstract must focus on the findings, and interpretation of the findings in the paper. The abstract must not make statements not addressed by the data presented within the paper. With this in mind, and my prior recommendation that the first two lines of the previous abstract be deleted, this stands as the most critically important item not accepted by the authors.
The first few lines of the Abstract leave the impression that data is presented within the paper that deals with the topic of whether reducing e-cigarette use will reduce teen smoking. Not only is no such data presented, the new text in the paper relative to the common liability hypothesis (lines 245 to 251) conflicts with the conclusion that this paper provides evidence that e-cigarette use leads some teens to smoking.
In addition, as a matter of principle, one cannot draw conclusions regarding cause and effect from cross-sectional data. This is a cross-sectional study. While the authors may be passionate about their belief that e-cigarettes lead teens to smoking, these first few lines must be deleted from the Abstract.
If the authors wish to deal with the topic of whether e-cigarettes lead to smoking, the only statement they can reasonably make is to the effect that more research will be required to determine whether reducing e-cigarette use will reduce teen smoking in Poland. Such a statement could be included in the paper, but would be inappropriate for the Abstract, no matter how passionate the authors are about this issue.
I therefore recommend that the first two and a half lines of the abstract be deleted, and that the abstract begin with the words “Teen use of tobacco related products is a substantial public health concern.” If the first two and a half lines are not removed, in the opinion of this reviewer, this paper should not be published.
In addition, in lines 39 - 41 of the abstract, the words “parental smoking and opinion of lower harmfulness of e-cigarettes use are the most important determinants of smoking susceptibility among never or ever e- cigarette users.” This should be replaced with words to the effect that the major determinant of e-cigarette use is cigarette use, referencing the other variables as significant but less important. (Please note that the use of the word “determinant” does not necessarily imply a cause/effect relationship).
The most important finding in the paper is “. Our results show that the determinants of e-cigarette use and cigarette smoking are the same.” Lines 344-345. This needs to be highlighted in the Abstract, not ignored.
Now to the issue of English grammar and word use. In this paper, authors use the terms “susceptibility,” “initiation,” and “use” as if they are synonymous. They are not. The terms “susceptibility” and “initiation” are used in a manner that implies a cause/effect relationship between e-cigarettes and smoking – an issue that the data presented in this paper do not and cannot address. “Use” is the proper word. In addition “e-cigarettes use” is grammatically incorrect. “e-cigarette use” is the proper wording.
In my most recent prior review of this paper, I urged that use of e-cigarettes does not constitute smoking. It constitutes e-cigarette use or vaping, but not smoking. This has been corrected in all but one spot – lines 208 and 209. Here the wording should be changed to say that most of the student’s parents did not use tobacco products.
I really would like this paper to be published, but the items referenced above must be corrected as a condition for publication.
Round 2
Reviewer 1 Report
Thank you for considering many of my minor comments. However, I still maintain my major concerns regarding the aims, results, and conclusions of this manuscript.
The aim of the study is much clearer and consistently defined, although it does not meet with the results. E-cigarette use refers to the current use status of a product, e.g. never/ever/current user while susceptibility to e-cigarette use means the lack of a firm commitment not to use a product in the future. Considering these definitions, the outcome variable in the present study (susceptibility to e-cigarette use) is incorrectly described at measures and throughout the manuscript and was called as “e-cigarette use” variable (Lines 150-157). Therefore, the whole text and of the results should be revised.
Clear data on the susceptibility to e-cigarette use is still lacking in the manuscript.
In lines 55-56, despite my previous review comment, the citation is still missing for this strong statement. For a similar statement in the Discussion part (Lines 310-313), cited papers do not support this statement.
As I previously noted in my review, “traditional cigarette” or “conventional cigarette” are still interchangeably used throughout the manuscript, although the Authors stated that they have changed it.
Regarding “type of school” variable in Table 2 and 3, the Authors have changed the reference category in Table 3 compared to Table 2 and gave no explanation for this. I have noted this in my previous reviews but still did not get any response for this.
The first sentence of the discussion was incorrectly amended. Instead of listing possible predictor variables, describing the sample of interest would be needed. That is, predictors of susceptibility to e-cigarette use among never and ever cigarette smoker adolescents…etc.
In the Discussion, lines 256-261, I cannot accept the explanation given for „the determinants of e-cigarette use are similar to the determinants of cigarette use”. The sample of the present study included adolescents, while in the recently added explanation the authors refers to an adult sample which is an incorrect way of comparison.
In lines 293-295, I cannot accept this explanation for the different results. It should have been explained more thoughtfully.
I still not agree with the major conclusion of this study, that is “Our results shown that the determinants of e-cigarette use and cigarette smoking are the same.” Such analyses were not included in the manuscript. Only the determinants of e-cigarette use susceptibility.
The English language improvement is still needed as numerous grammatical and writing mistakes exists in the text.
Reviewer 2 Report
-- so close. The authors are to be commended in making the changes needed in the Abstract and the entire paper, with the exception of the last three lines of the paper.
As I have repeatedly said before, no matter how passionate the authors are about the issues of harm reduction and gateway, they must not draw conclusions on these issues unless such conclusions are strongly supported by the data and narrative within the paper.
These new last three lines draw conclusions not justified by either the literature reviewed or the data and narrative presented in the paper. Here I refer to the two sentences beginning with "At the same time . . ." on line 352 and ending with "also needed" on line 354.
Nothing in this paper or the material reviewed deals with the issue of whether the difference in harm, comparing e-cigarettes to cigarettes, rules out the possibility that switching to e-cigarettes can be considered effective harm reduction. "Harm reduction" means "less harm," not "zero harm."
By the same token, the literature showing e-cigarette use prior to cigarette use does not constitute a gateway effect, unless one can demonstrate that eliminating e-cigarettes will reduce cigarette smoking by youth. References 35 and 36 suggest the possibility of a gateway effect but do not rule out common liability as the cause of this finding.
I therefore suggest that the authors choose between one of three options to correct this serious error in conclusion:
1. Replace this statement with a statement from a prior draft of the conclusion "At the same time, they should also take into account the harmfulness of traditional cigarettes, so as not to direct vapers and potential vapers to cigarettes (and to marijuana).
2. Refer to harm reduction and the gateway issue as items in need of additional research.
3. Simply delete this statement and end the conclusion at the first word in line 352 "products."